# Refined Reconstruction of Global Prostate Segmentation from Patch-wise Coarse Predictions

**Jian Kang**                                                                JIAN.KANG@STUDENT.UNSW.EDU.AU

**Gihan Samarasinghe**                                       GIHAN.SAMARASINGHE@UNSW.EDU.AU

**Shuchao Pang**                                                       SHUCHAO.PANG@UNSW.EDU.AU

**Arcot Sowmya**                                                                     A.SOWMYA@UNSW.EDU.AU

*School of Computer Science and Engineering, University of New South Wales, Sydney NSW 2052, Australia*

**Editors:** Under Review for MIDL 2021

## Abstract

Whole organ segmentation in biomedical images continues to be an important problem. Recently deep learning based methods have produced convincing results. Taking into account limited sample sizes and inconsistent spacing on devices, many solutions are designed to work patch-wise, with networks trained on sub-volumes with a certain stride, rather than on the whole 3D image. The binarized segmentation probability map of the whole volume image is then computed by mapping back stacked sub-volume predictions using a threshold. As may be expected, the performance is highly sensitive to the threshold chosen, as well as issues such as low probabilities on object boundary region, missing voxels or distortions on corners of sliding windows and unexpected components. In this work, we analyse and test the performance of a commonly used thresholding method, and introduce learning based methods to address these issues. In addition, a simple shape prior from a size-based shape heat map is introduced to improve overall performance. Experiments were carried out on the open MICCAI PROMISE12 challenge dataset for prostate segmentation. With the learning based method, the average performance increased by 1.65% on Dice Similarity Coefficient(DSC) compared to the thresholding method. On challenging cases, the improvement was more than 6.55% on DSC. Since proposed method produces convincing results with only modifications of the reconstruction step, this approach may be adopted in other patch-wise deep learning networks too.

**Keywords:** patch-wise prediction, deep learning, prostate segmentation, reconstruction

## 1. Introduction

Accurate prostate segmentation enables better visualization and localization of suspicious lesions, which helps in diagnosis, treatment planning and disease prognosis. Most prostate segmentations are manually annotated by physicians slice by slice, which is tedious, time-consuming, and subject to inter- and intra-reader variations. Many attempts(Liu et al., 2019; Lei et al., 2019; Liu et al., 2020; Gillespie et al., 2020)have been made to achieve automatic segmentation of the prostate. In biomedical image segmentation, encoder-decoder structures are widely utilized, from the simple Unet(Ronneberger et al., 2015) to the more complex Resnet(Yu et al., 2017) and Densenet(Yuan et al., 2019). Modifications and extensions, such as Znet(Zhang et al., 2019) and cascade dense Unet(Li et al., 2019) have

been developed to improve the overall performance. Moreover, shape models are combined with deep learning architectures(Cheng et al., 2016) to enhance global features. However, the necessary sample size increases with network complexity. Generally this problem is addressed by training the network on smaller patches and adding more samples with data augmentation methods. Taking into account the limited sample sizes and inconsistency of spacing from different devices, many models are designed to work patch-wise(Qin, 2019; Jia et al., 2019). Whenever a new test sample is to be predicted in a patch-wise deep learning network, four steps are required, as shown in Figure 1. Firstly, the original input is scanned and divided into several sub-volumes with a certain stride. Then predictions on the sub-volume images are generated from a well-trained deep learning network. After that, the sub-volume predictions are mapped back to reconstruct an overall prediction for the test data. Finally the overall prediction is binarized to produce the final output for segmentation prediction. Commonly the binarization is performed by using a thresholding method, which is highly sensitive to the threshold value chosen. Since the overall prediction is obtained by binarizing the reconstruction of overlapping small patches, it may be affected by issues such as low probability on object boundary region, missing or distorted part on corners of the sliding window and unexpected particles.

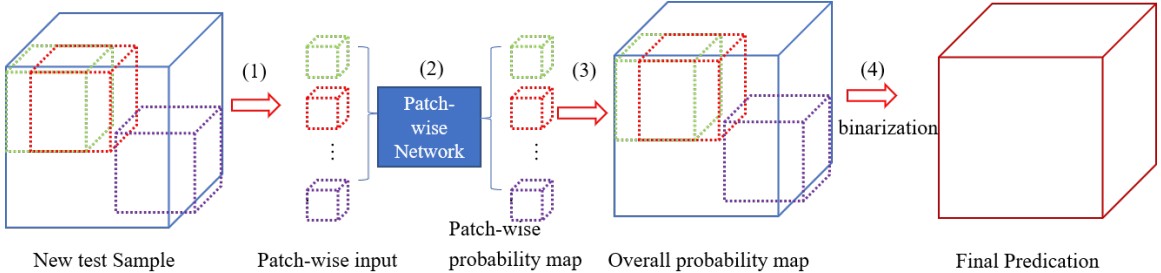

Figure 1: The workflow of predicting a new test sample in patch-wise deep learning networks

To deal with these issues, this work is aimed at finding a more robust method to handle the binarization of a reconstructed probability map from a patch-wise deep learning network. Our main contributions are threefold:

1. the first discussion of binarization issues in the reconstruction step of patch-wise deep learning networks with thresholding method;

2. proposal of a learning-based method to deal with the binarization procedure as a coarse to fine problem;

3. introduction of a simple size-based shape prior to improving overall performance.

## 2. Methods

This research is based on the output from a 3D Unet trained on the MICCAI PROMISE12 challenge dataset(Litjens et al., 2014) with a small patch size of 96*96*32. In the inference

phase, each MR image is scanned with sub-volumes of the same size (96*96*32) and a fixed stride of 32*32*8. The overall probability map is acquired by normalizing the summation of the patches to 0 ~1. Our work focuses on analysis and improvement of the issues encountered in the reconstruction step.

## 2.1. Problem Statement

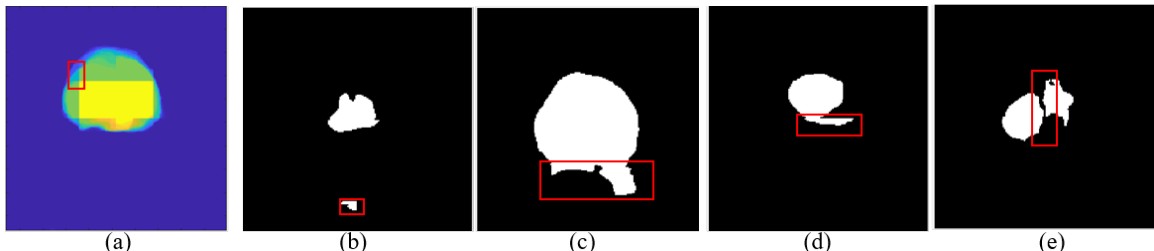

| (a) | (b) | (c) | (d) | (e) |

Figure 2: Several samples of predictions from the thresholding method: (a) one slice from overall reconstructed probability map (the brighter the more likely to be the target); (b) unexpected particles; (c) distortion of boundary region; (d) distortion from sliding window corner; and (e) missing and isolated parts.

Generally, a binarized output volume is simply acquired using a threshold T, however this may give rise to some issues. In Figure 2, some bad cases of binarization with threshold T=0.3 are shown. In Figure 2(b), an unexpected small particle is visible at the bottom of the image. In Figure 2(c), distortions in boundary regions with a higher probability value than the threshold are shown. In Figure 2(d) the distorted part is caused by the sliding window. In Figure 2(e), the missing parts that have a smaller probability value than the threshold are demonstrated. There are mainly four reasons why the results are sensitive to the threshold chosen, which may be ascribed to the incoherence of the probability values for voxels. First, the probability values for challenging cases are much lower than the more normal common cases. Secondly, the voxel probability values on boundary regions(light blue region) are much lower than in the centre zones(yellow region) as shown in Figure 2(a). Thirdly, the patch-wise network may ignore the target(red rectangle in Figure 2(a)), if it is located at the corner of the sliding window and occupies only a small part of the sliding window. Fourth, some unexpected small particles with local features similar to the target object may be mistakenly judged as the target.

Table 1: Results of Dice similarity coefficient on several cases under different thresholds

| T | 0.1 | 0.2 | 0.3 | 0.4 | T | 0.1 | 0.2 | 0.3 | 0.4 |
|---|---|---|---|---|---|---|---|---|---|
| Case8 | 79.70 | 88..07 | 88.97 | 83.65 | Case18 | 87.04 | 85.81 | 83.68 | 79.92 |
| Case23 | 85.10 | 79.31 | 72.71 | 70.66 | Case31 | 79.14 | 85.43 | 85.77 | 88.67 |
| Case35 | 87.85 | 88.52 | 88.62 | 87.27 | Case46 | 81.89 | 84.27 | 85.75 | 86.72 |

In Table 1 the DSC changes with different Threshold values on 6 cases are shown, which implies that the performance is highly sensitive to the threshold T. Observing that the best threshold value differs case by case, we tested the data on the Otsu Threholding method(Yousefi, 2011) which is self adaptive. Even though the adaptive threshold method handles probability incoherence from different cases, it is unable to deal with the problems caused by sliding windows and unexpected particles caused by patch-wise learning.

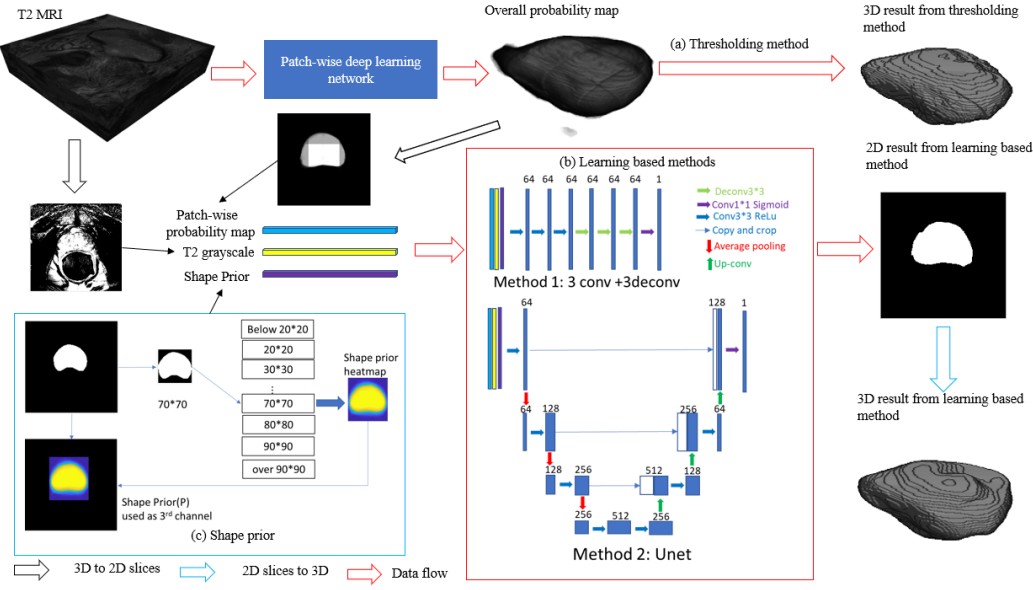

Figure 3: Our proposed learning-based method for refined reconstruction of global prostate segmentation: (a) the thresholding method; (b) the learning-based binarization method; and (c) the extraction of shape prior as parts of inputs in this study.

## 2.2. The Learning-based Method

Since a simple thresholding method is unable to deal with the binarization problem efficiently, learning-based methods are introduced in our work. Assuming the patch-wise probability to be a coarse prediction, the problem to be solved can be seen as a coarse to fine model. To avoid the patch-wise issues mentioned in section 2.1, learning based methods are designed to work globally, therefore the whole target will be included as input. With limited sample number in the 3D domain, we trained our networks on 2D slices. To keep the input size coherent, an ROI of size 192*192 is extracted from the image center. A light-weight network inspired by ShapeMask(Kuo et al., 2019) and a 2D Unet with average pooling are tested separately. In order to make full use of the data, the original T2 MRI image is deployed as another channel, and a size-based shape prior is introduced as the third channel to improve the overall performance. The learning based methods are shown as Figure 3. The T2 MR image and overall probability map are downsampled to 2D slices along the axial axis. Then a shape prior is used as the third channel. Therefore, each input

is a 2D image of size 192*192 and the 3 channels(192*192*3) include the T2 image, the reconstructed probability map and the shape prior. For Method 1, a combination of conv and deconv layers with limited feature maps are used. For Method 2, a widely used Unet with average pooling instead of Max pooling is used. Finally, the 3D result is created by stacking the 2D slice results from a learning based method.

### 2.3. Shape Prior

Common encoder-decoder networks focus more on local features. Many efforts have been made to add shape information to encoder-decoder networks. Mirikharaji et al.(Mirikharaji and Hamarneh, 2018) introduced the star shape prior into a common CNN to improve performance. An auto-encoder is used to combine the CNN prediction and anatomical shape prior in other work(Oktay et al., 2017). In our case, a coarse prediction is obtained from a patch-wise deep learning network, which helps to fix an approximate size and position. Besides, the size and shape of the prostate differs considerably at the top, middle and lower parts of the organ.Taking these properties into account, a size-based atlas set would be appropriate. The ground truth for all patients in the training set is checked slice by slice to decide on the number and range of the subsets,including below 20,20*20,30*30...90*90 and above 90. Afterwards the Shape Prior heat-map is obtained by taking the average of all slices of the particular size, 70*70 for instance in Figure 3(c), which shows how to obtain a shape prior as the third channel of input for a new sample. First, we extract the centre and size of the coarse prediction. Afterwards the original image is replaced by a shape prior heat-map corresponding to its centre and size.

## 3. Experimental Results and Discussion

The experiments were carried out on the MICCAI PROMISE12 challenge dataset. The training dataset contains 50 transversal T2-weighted MR images of the prostate and the corresponding segmentation ground truth. The test dataset consists of 30 MR images, however the ground truth is held by the organiser for independent evaluation. Since the images were acquired from four different hospitals, using different devices and different acquisition protocols, there are large variations in voxel intensity, dynamic range, position, field of view and anatomical appearance. Therefore some simple preprocessing was applied to the original data, including voxel spacing unification to a fixed size of 0.625*0.625*1.5mm and intensity normalization into zero mean and unit variance. For visual comparison, in this work experiments were only carried out on the training dataset. Ten fold cross-validation was applied to generate training and test subsets for our experiments. For the patch-wise deep learning network, the widely used 3D Unet was trained with sub-volumes of size 96*96*32. In the inference phase, each MR image was scanned with sub-volumes of the same size (96*96*32) and a fixed stride of 32*32*8. Quantitative results from 10-fold cross validation are shown in upper part of Table 2. To further validate the effect of our proposed method on different patch-wise networks, the results from first fold among 10-folds using 3D Resnet(Yu et al., 2017) as another patch-wise deep learning network is shown in lower part of Table 2. The results were evaluated with Dice similarity Coefficient (DSC), Hausdoff Distance (HD) and absolute Relative Volume Difference (aRVD).

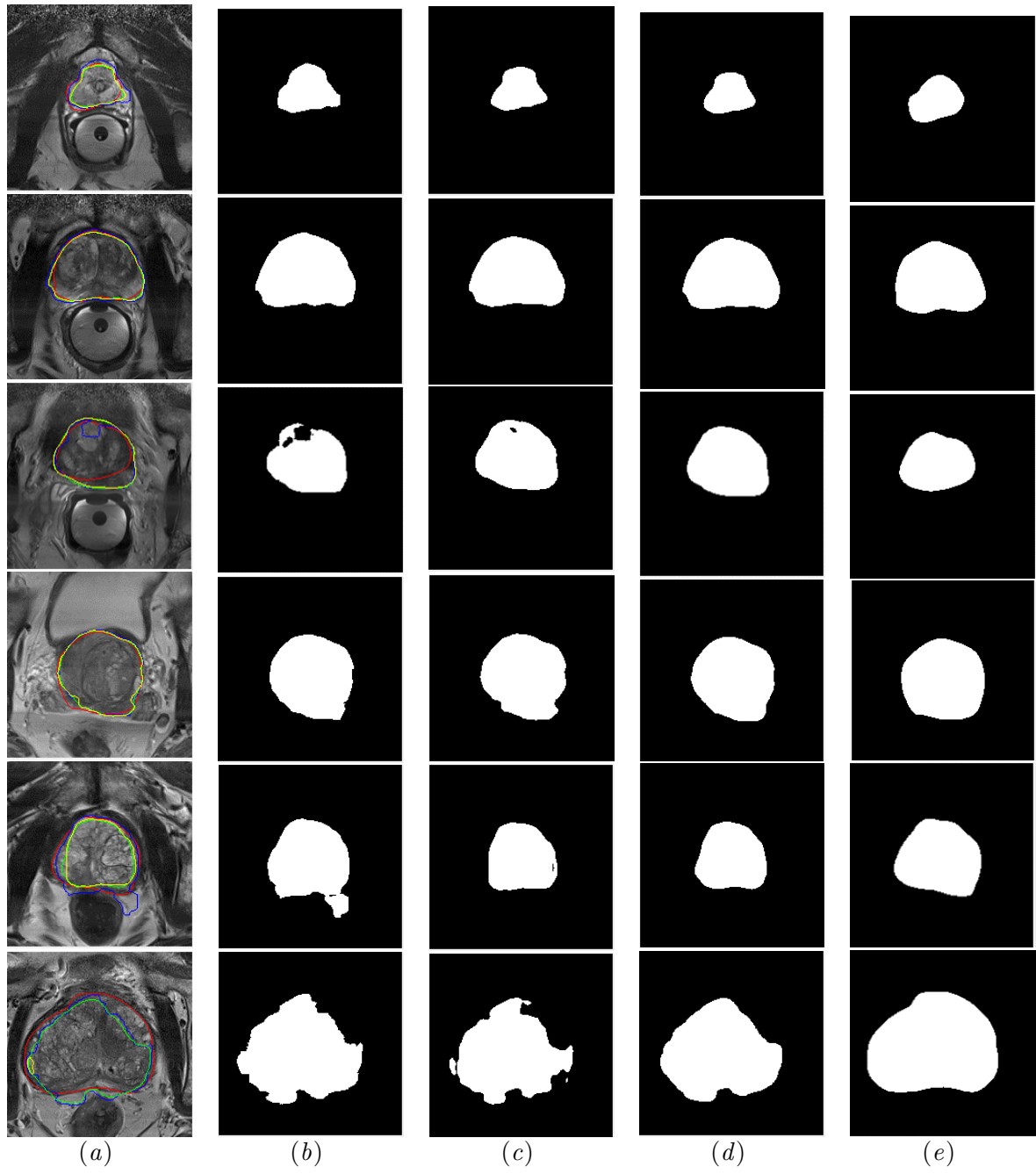

Figure 4: Qualitative comparison of different binarization methods: (a) different delineations of the prostate; (b) T=0.3(blue); (c) Method 1(yellow); (d) Method 2(green); and (e) Ground Truth(red).

Table 2: Quantitative comparisons of different binarization methods under widely used evaluation metrics using two patch-wise networks

| Patch-wise Network | Method | DSC(%) | HD(mm) | aRVD (%) |
|---|---|---|---|---|
| 3D Unet | T=0.2 | 85.44±13.66 | 12.15±13.48 | 16.24±18.74 |
| | T=0.3 | 86.57±13.03 | 9.34±8.72 | 13.13±13.70 |
| | T=0.4 | 84.79±14.89 | 11.26±10.33 | 13.97±14.14 |
| | Otsu | 86.04±12.67 | 9.75±7.52 | 13.07±8.34 |
| | Method 1 | 87.48±7.22 | 7.75±3.23 | 8.89±6.35 |
| | Method 2 | 88.22±6.83 | 6.52±2.87 | 7.12±6.02 |
| 3D Resnet | T=0.2 | 87.13±5.58 | 7.95±5.87 | 11.86±12.86 |
| | T=0.3 | 86.53±8.91 | 8.42±7.58 | 10.15±15.02 |
| | T=0.4 | 85.05±13.68 | 11.70±11.24 | 11.46±17. 94 |
| | Otsu | 87.35±5.59 | 7.34±4.12 | 8.61±10.86 |
| | Method 1 | 88.77±2.92 | 4.53±2.73 | 6.93±6.24 |
| | Method 2 | 89.01±2.72 | 4.36±2.45 | 5.45±4.68 |

From the results, we can see that the adaptive thresholding(Otsu) method does not always outperform the manually chosen best threshold, however it provides a relatively better solution than a randomly selected threshold. Among the learning based methods, Method 1 inspired by ShapeMask and Method 2 based on a deep learning network both improve the overall performance. Our proposed reconstruction framework, can be easily migrated to other patch-wise deep learning networks, such as Resnet.

In order to better evaluate the performance, we extracted 6 slices as shown in Figure 5. Among them, rows 1, 2 and 3 denote the top, middle and lower slices of one patient. Row 4 shows a common case on another patient, and rows 5, 6 indicate challenging slices. From the results, we can observe that both ShapeMask and 2D Unet networks improve the overall performance. While the ShapeMask based Method 1 attempts to sharpen the image edges, the 2D Unet network based Method 2 attempts to find a balance between sharpening and compensation, as shown in row 6 of Figure 4.

Table 3: Results of Dice similarity coefficient with different inputs

| Input Channels | P | P+ S | P+S+T |
|---|---|---|---|
| Method 1 | 86.84±12.87 | 87.23±11.32 | 87.48±7.22 |
| Method 2 | 87.12±10.58 | 87.58±11.27 | 88.22±6.83 |

To evaluate the contributions of the proposed shape prior, we tested the reconstruction system with different inputs, as shown in Table 3. The second column(P) shows the result of using only the coarse prediction as our input. The third column(P+S) indicates the result of adding Shape Prior as another channel of input. The inputs of the fourth column(P+S+T) cover all three channels, including coarse prediction, Shape Prior and T2 Image. From the

results, we can see that the newly proposed Shape prior helps to enhance the accuracy of the proposed methods.

In order to get a better understanding of the problem, histograms of probability map are extracted from the patch-wise network, as shown in Figure 5. For easy cases, such as Case 14 and Case 22, most voxels are distributed on both sides and their ratios of uncertain voxels in the threshold sensitive region (red rectangle) are quite low. For Case 34, the threshold can be shifted to the valley with Otsu thresholding method to improve the overall performance. However, when it comes to the challenging Case 23, the problem cannot be handled with only a thresholding method. Instead, our learning based method, which includes global features and shape prior, helps to significantly improve the accuracy on Case 23. Furthermore, Table 4 shows the DSC changes of the four cases listed in Figure 5. For challenging cases such as Case 16, Case 37 and Case 23, our proposed method dramatically improves the performance by more than 6.55% on DSC for each case.

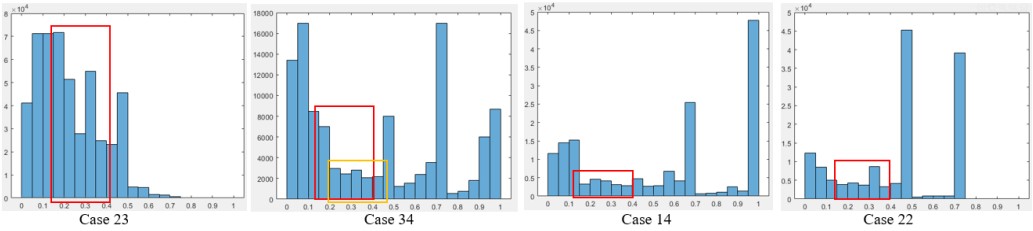

Figure 5: Histograms of voxel probability map on several test cases

Table 4: Results of Dice similarity coefficient on particular cases with different methods

| Method | Case16 | Case37 | Case23 | Case34 | Case14 | Case22 |
|---|---|---|---|---|---|---|
| T=0.3 | 80.79 | 82.44 | 72.71 | 86.63 | 89.47 | 89.37 |
| Otsu | 80.53 | 81.88 | 68.41 | 89.14 | 88.65 | 88.86 |
| Method 1 | 87.16 | 86.71 | 82.39 | 86.85 | 92.42 | 89.30 |
| Method 2 | 87.45 | 88.99 | 85.93 | 86.73 | 92.48 | 89.13 |

## 4. Conclusion

In this paper, we have addressed the binarization issues with thresholding methods in the reconstruction step of patch-wise deep learning networks. Then learning based methods, including Method 1 based on ShapeMask and Method 2 based on Unet, are introduced to deal with the binarization procedure as a coarse to fine problem. Besides this, a size-based shape prior is proposed to improve overall performance. Experimental results demonstrate that our proposed methods further improve the performance of different patch-wise deep learning networks for prostate segmentation. Since the refinement is applied only at the reconstruction step, it can be easily migrated to other patch-wise deep learning networks.

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
