# OpenReview forum: "Refined Reconstruction of Global Prostate Segmentation from Patch-wise Coarse Predictions"
_MIDL.io/2021/Conference — Submitted to MIDL 2021_

### Official Review · AnonReviewer3 · 2021-03-04

**Confidence:** 4
**Preliminary Rating:** 1
**Final Rating:** 1

**Summary:**

The paper investigates the problem of aggregating patches back to entire image when using patch-based segmentation on the problem of prostate MR segmentation. First, the paper shows the problem of using thresholding of the probability maps resulting in false positives and false negatives. Second, a CNN-based method is introduce to alleviate the problem of patch aggregation. The CNN-based method improves the segmentation performance over the thresholding of the probability maps.

**Strengths:**

-	Ablation of the proposed solutions in Table 3
-	Three metrics, considering different aspects like overlap, distance, and volume, are used for the evaluation. The method is evaluated on a public dataset.
-	Good illustration of why the thresholding does not work for challenging cases (Figure 5); could actually be moved to Sec 2.1
-	Nice Figure 4 that visually shows the differences between the methods


**Weaknesses:**

-	The paper lacks proper references to existing work specifically to the problem of patch-wise reconstruction. E.g., the MICCAI 2020 paper by Madesta and Schmitz et al. investigates patch sampling and inference strategies. Also, Audelan et al., MICCAI 2020 might be relevant for the work at hand. Surely, there is other relevant literature on patch-based reconstruction, and the literature review should likely go beyond prostate segmentation, as the main contribution of the paper is the patch-wise reconstruction (e.g., have a look at the references of the provided papers)!
-	I am not sure if the statement “taking into account (…) inconsistency of spacing from different devices, many models are designed to work patch-wise” is correct. How does a patch-wise approach solve inconsistencies in spacings? Also, the references Qin 2019 and Jia 2019 do indeed work patch-wise, however the spacing is not mentioned as a reason. Rather, I think it is the limiting GPU memory. Heterogenous spacings are usually just homogenised by resampling, as the authors also chose to do.
-	It is unclear what 3D UNet the work used to obtain the patches (Sec 2, first sentence). Why does table 1 only show results from six cases, is this the testing set? How is the performance over the entire testing set, the table looks like the results are cherry picked. What about the variability of the results? I would prefer to have a box plot with a box for each threshold to better acknowledge the influence of the threshold. Otsu could also be included in such a box plot. Table 2 reports results over the entire data set (I believe), so what is the point of table 1?
-	The description of the learning-based method could be improved. What exactly is the motivation? I think the method is 2D because of memory issues, right? Please describe, otherwise, the motivation is unclear. Re-using the probability map is heavily inspired by Auto-context (Mohseni Salehi et al., IEEE TMI, 2017), please reference appropriately. The shape prior is essentially an atlas, right? Why is the shape prior not constructed in 3D but slice-wise? Further, there exists literature closer to your approach than the referenced Oktay at al., which use an atlas as shape prior, e.g., Zotti et al, 2018, IEEE Bio and Health Informatics. Finally, as the method is 2D, it might be good to have a simple 2D Unet as an additional baseline to the thresholds and Otsu. Or in other words, does the 3D patch-wise processing really help or is 2D enough?
-	The method is not reproducible as public code is not available.
- The cross-validation is not entirely clear. Was the method only tuned on one fold in the CV? The 10-fold seems to be resulting in a very small validation set of 5 cases, and tuning on one fold only might result in overfitting on the 5 cases of this fold. So, I am wondering if more than one fold were used for tuning? A 5-fold CV seems more appropriate (and would also reduce the computational time).
-	Overall, the results seem not to approach state-of-the-art, see the leader board available at https://promise12.grand-challenge.org/evaluation/challenge/leaderboard/ E.g., the nnUNet by Isensee et al. achieves a Dice coefficient of around 0.896. The patch-size of the nnUNet also hints that the solution to the problem the paper tackles might simply be a different patch size of 28 x 256 x 256 (see https://arxiv.org/pdf/1904.08128.pdf) rather than 32 x 96 x 96, which would provide more 2D context while sacrificing context in the z-axis. Therefore, the proposed learning-based solution might become obsolete.
- There is a lack of baselines in the paper that puts the result into the context of the challenge submissions.  What is the performance of the method on the challenge test set? The authors could submit to the challenge and obtain a rank as a paper has been written (see guidelines of the challenge).


**Deanonymize Review:**

no

**Detailed Comments:**

Fig 2 and Sec 2.1:
-	I like the idea to visualize the problem you are tackling by such a figure with explanation in the main text
-	Please add a colorbar to the probability map
-	it is unclear, how you arrive from a) to b), c), d), and e). I assume it is not the same case?
-	Maybe adding the structural MR image might be helpful

Others:
-	Abstract: “and unexpected components”: unclear what is meant by this statement. Maybe over- and under-segmentation or false positives and negatives?
-	Spaces are often missing before and after references and around parenthesis.
-	Introduction: “unexpected particles”: what is meant by this statement?
-	Introduction: “the first discussion”, are you entirely sure? I am sure you are not the first, see Madesta and Schmitz et al., MICCAI 2020
-	Sec 3: there is a mix of methods and results in the first paragraph.
-	Sec 3: was the method only tuned on one fold in the CV?
-	Overall: Please proof-read the paper; there exist several grammatical mistakes that, when fixed, would improve the reading experience


**Final Rating Justification:**

The authors did not respond and it would anyway have been very difficult to change my mind regarding the rating. Therefore, I keep my rating.

**Justification Of The Preliminary Rating:**

I consider the changes that need to be addressed in the rebuttal as a major revision. I don’t think that the time is enough to address my concerns appropriately and, therefore, recommend rejection of the paper.

**Paper Type:**

both

**Questions To Address In The Rebuttal:**

There is too much to address for a rebuttal (see weaknesses), including large portions of the paper to be re-written.

**Special Issue:**

no

---

### Official Review · ~Hans_Meine1 · 2021-03-05

**Confidence:** 5
**Preliminary Rating:** 1
**Final Rating:** 1

**Summary:**

The authors investigate how to deal with problems resulting from using a 3D U-net with "same" padding and a small patch size, applied with overlapping tiles. The manuscript clearly describes the resulting problems (in conjunction with a threshold of 0.3). Unfortunately, the whole presumption (hope that's the right term) that these are common problems that need to be addressed is false; proper overlapping tile strategy (cf. original U-net paper), large patches, or calibrated models would fix the root cause of these observations. Hence, the manuscript does not really contribute to the state of the art.

**Strengths:**

The paper illustrations are helpful, the tables nicely formatted, and the text is reasonably structured and readable. It's easy to follow the thoughts of the authors, even if one does not share them.

The authors at least perform one ablation study (albeit not very complete) w.r.t. the combination of multiple channels for the postprocessing (Table 3, likely referring to the 2D Unet although I think it is not stated).

**Weaknesses:**

The authors base their work on false assumptions on the relevance and cause of the problems they observe.

The validation does not include statistical significance analyses. The data behind tables 2 and 3 could have better been summarized with boxplots, I think. The text or captions do not seem to state which statistics are applied; I assume the numbers are mean and stddev., which are suitable measures for normal distributions, but normality may not be given here.

The choice of the networks appears to be a little arbitrary, a discussion of the rationale, the relevant differences, and possibly more ablation studies are missing.

**Deanonymize Review:**

yes

**Detailed Comments:**

If you have purposely used such a small patch size to exaggerate the problems, it would be good to state so. Even with same padding, there are certainly better results possible. (Cf. nnU-net)

The histogram in Fig. 5 shows a clear pattern where there are many regularly spaced peaks – this stems from the overly confident output of (typically softmax-using) CNNs, see literature on model calibration, and the averaging of multiple overlapping predictions. In this situation it is clear that the Otsu method (which expects a bimodal distribution) should not be applied.

When reading "from the results, we can see that", it was not clear to me whether the reader should be able to see that himself/herself too, and how. Ideally, one would draw such conclusions from appropriate statistical testing.

I would not call Resnet "more complex" than Unet, and the cited work by Lu et al. should not be referred to as just "Resnet" (which would be universally interpreted as the work by Kaiming He).

The formatting around punctuation is strange; often there are spaces missing around parentheses or sometimes after a full stop. "Threshold" is wrongly capitalized at least once.

I fail to see how shape models "enhance global features". (Maybe the authors meant "contribute more global features" or so?)

**Final Rating Justification:**

No rebuttal, no revision, other reviewers basically agreed, so I keep my previous rating.

**Justification Of The Preliminary Rating:**

Unfortunately, as argued above, I think this paper does not bring any significant contribution. The problems the paper tries to address are clearly caused by a bad inference strategy (and possibly too small patches during training as well), and the methods proposed to address them do not fix the root cause, are itself based on bad assumptions (cf. Otsu), and are not well justified.

**Paper Type:**

methodological development

**Special Issue:**

no

---

### Official Review · AnonReviewer4 · 2021-03-06

**Confidence:** 4
**Preliminary Rating:** 2
**Final Rating:** 2

**Summary:**

This work proposes to use learning to solve the problem of stacking 3D prediction patches into a full prediction volume, a problem that is commonly solved by just stacking the patches and using thresholds. Instead, the authors use 2D networks to compute the final segmentation from the 3D patch predictions, including shape priors on the input.


**Strengths:**

The proposed solution of using 2D networks to reconstruct patch-wise results from 3D networks is interesting, especially in the inclusion of shape priors to guide the reconstruction. Figures are well made in presenting the problem and showcasing the proposed method.


**Weaknesses:**

One major point of comparison that is lacking is: what would be the performance of simply training the learning method on the input data and stacking 2D predictions into a volumetric prediction? This could be presented as using only T as input in table 3. In this way it would be clear that using the 3D predictions is really advantageous.

The authors do not address the network size increase of adopting a full 2D UNet as a reconstructor of the 3D output. I wonder if increasing the original 3D UNet by a similar number of parameters would lead to a similar increase in performance even using the traditional thresholding method?

From the manuscript it seems the learning method was trained separately. Information on the additional time needed in relation to only training the original 3D network is necessary.

There are some writing mistakes and typos that could be easily corrected with proofreading. The introduction initially focuses on prostate segmentation, however the main point of the paper is proposing a better way to stack volumetric predictions than thresholding.

The high standard deviation on Table 2 raises concerns that the results might not be statistically significant, although the reduction on standard deviation showcases more stable segmentations when using the learning methods.

Finally, providing or commiting to provide implementation code when published could improve the paper strength and reproducibility.


**Deanonymize Review:**

no

**Detailed Comments:**

Abstract last phrase is missing an article: “Since the proposed method[...]”

The paper has some wrong spacing on parenthesis.

I would avoid using too many adjectives, e.g categorizing UNet or Resnet as simple or complex.

Figure 2 a) could have a legend mapping colors to predicted value, instead of the caption sentence “(the brighter the more likely to be the target)”. The caption should reference the red rectangle.

It should be clearer in Figure 3 that the Learning Based method consists of 2D networks. It is also not clear how the patch wise predictions are brought back to the volume shape to be used as input in the learning based method, since the 32*32*8 stride brings overlap between patches.

All tables could use a highlight of the higher/better metrics. Table 3 comparison with different inputs is excellent, why not also adding P+T results?


**Final Rating Justification:**

There was no rebuttal so I stand with my initial rating.

**Justification Of The Preliminary Rating:**

The idea of using learning to reconstruct from 3D patch predictions is interesting and seems to bring positive results. However, the study could be more complete to really show that both the 3D and 2D networks are necessary to the achieved results, and that they are statistically significant. Author’s responses to my inquiries and clarifications might improve the rating.


**Paper Type:**

both

**Questions To Address In The Rebuttal:**

The authors are welcome to address the points I raised in previous sections, as they are separated in a paragraph for each point.

**Special Issue:**

no

---

### Official Review · AnonReviewer2 · 2021-03-09

**Confidence:** 4
**Preliminary Rating:** 2
**Final Rating:** 1

**Summary:**

This paper proposes a method to regularize the prediction of a 3D CNN, to improve the quality of the final segmentation.

The proposed regularizer takes the form of a trainable 2D Unet, coupled with the existing star shape prior. The method is evaluated on a single dataset, PROMISE12 (prostate segmentation).

**Strengths:**

- The authors report several metrics
- The dataset, PROMISE12, is challenging, and is useful to benchmark different methods.
- The topic of segmentation regularization/post-processing is important and relevant

**Weaknesses:**

I am not even sure that I understood the problem that the authors attempt to solve. Usually binarization is done simply by training on two classes (background and foreground), and performing an argmax to get the final prediction. Temperature might help at the softmax level, but usually it is not required (especially on PROMISE12), and finding an optimal thresholding value becomes irrelevant.

Moreover, the paper is not very clear in what exactly their baseline does. The authors mention:
> The overall probability map is acquired by normalizing the summation of the patches to 0 ~ 1

This implies that one needs to compute all overlapping patches to predict a single pixel probability. Therefore, I am asking, how do you train each sub-patch, if you need to output from the other sub-patches to compute it?

Why not simply predicting two logits per pixel in each sub-patch, and performing a softmax on it? This would allow to train each patch independently. Then, when reconstructing the final 3D volume, one could merge the overlapping predictions by averaging them. If this method poses an issue, then you would have an actual problem to solve. But right now, I see this paper as trying to fix a non-issue, that arose from a sub-optimal implementation.



The topic of post-processing/regularizing the predicted segmentation is important. However, several important baselines are missing:
- the proposed regularization method is a simple 2D UNet. Therefore, another baseline (3D segmentation as a series of 2D segmentation, with UNet), is required. I know such methods can get very close to 90% DSC on PROMISE12 quite easily
- Computational cost between different methods is needed, and a discussion on the trade-offs would be very welcome
- Adding a 3D graphcut or 3D CRF post-processing baseline is needed, as the proposed method mostly smooth-out the contour of the predicted segmentation


Overall, the writing and the clarity of the paper could be improved, it is a bit hard to follow at times. For instance, replacing "method 1" and "method 2" with a high-level description of the method would streamline the reading of the tables a lot.

**Deanonymize Review:**

no

**Detailed Comments:**

Minor:
- Figure 2 requires a color scale
- A space is required between parenthesis and the text. This also applied for citations.

**Final Rating Justification:**



**Justification Of The Preliminary Rating:**

I am puzzled by this paper, I am not even sure I understood the problem the authors face. A lot of clarification will be required during the rebuttal.

If I understood correctly, then the evaluation is lacking in several aspects, with missing baselines (simple 3D segmentation as a series of 2D segmentations, graphcut/CRF methods comparison).

**Paper Type:**

methodological development

**Questions To Address In The Rebuttal:**

- Can you clarify _exactly_ how the patches are trained, their probabilities computed, and how the final 3D volume is reconstructed?
- Can your method be extended to a true multi-class setting (more than foreground and background)?
- Due to the use of the star-shape prior, would your method generalize to tasks involving multiple connected components?

**Special Issue:**

no

---

### Meta-Review · Area_Chair1 · 2021-03-22

**Recommendation:** Reject

**Metareview:**

The reviewers have provided very detailed, argumented and constructive criticism of the paper.

Although they acknowledge some interest to the paper, they all agree on the fact that this paper had major flaws to be adressed (missing baselines, lack of clarity, code availability).

The authors did not provide a rebuttal.

I follow the reviewer’s rating and recommend rejection of this paper.

**Paper Type:**

both

---

### Decision · Program_Chairs · 2021-03-31

Reject